# Protecting Physical Layer Secret Key Generation from Active Attacks

**DOI:** 10.3390/e23080960

**Published:** 2021-07-27

**Authors:** Miroslav Mitev, Arsenia Chorti, E. Veronica Belmega, H. Vincent Poor

**Affiliations:** 1Barkhausen Institut gGmbH, Würzburger Str. 46, 01187 Dresden, Germany; 2ETIS, UMR 8051 CY Cergy Paris Université, ENSEA, CNRS, 95000 Cergy, France; arsenia.chorti@ensea.fr (A.C.); veronica.belmega@ensea.fr (E.V.B.); 3School of Engineering and Applied Science, Princeton University, Princeton, NJ 08544, USA; poor@princeton.edu

**Keywords:** physical layer security, secret key generation, injection attacks, jamming attacks, pilot randomization

## Abstract

Lightweight session key agreement schemes are expected to play a central role in building Internet of things (IoT) security in sixth-generation (6G) networks. A well-established approach deriving from the physical layer is a secret key generation (SKG) from shared randomness (in the form of wireless fading coefficients). However, although practical, SKG schemes have been shown to be vulnerable to active attacks over the initial “advantage distillation” phase, throughout which estimates of the fading coefficients are obtained at the legitimate users. In fact, by injecting carefully designed signals during this phase, a man-in-the-middle (MiM) attack could manipulate and control part of the reconciled bits and thus render SKG vulnerable to brute force attacks. Alternatively, a denial of service attack can be mounted by a reactive jammer. In this paper, we investigate the impact of injection and jamming attacks during the advantage distillation in a multiple-input–multiple-output (MIMO) system. First, we show that a MiM attack can be mounted as long as the attacker has one extra antenna with respect to the legitimate users, and we propose a pilot randomization scheme that allows the legitimate users to successfully reduce the injection attack to a less harmful jamming attack. Secondly, by taking a game-theoretic approach we evaluate the optimal strategies available to the legitimate users in the presence of reactive jammers.

## 1. Introduction

The increasing interest in physical layer security (PLS) has been stimulated by many practical needs, particularly in the context of Internet of things (IoT) applications [1]. For example, in [2,3], secret key generation (SKG) from wireless fading coefficients was analyzed, showing its potential as a lightweight alternative to standard security schemes. In fact, the SKG scheme allows two legitimate parties (Alice and Bob) to extract on-the-fly secret keys, without the need for significant infrastructure. Furthermore, it has been information-theoretically proven that by following the SKG process, Alice and Bob can extract a shared secret over unauthenticated channels [4,5,6]. Building on that, numerous practical experiments have demonstrated the feasibility of the scheme [7,8]. Moreover, it has been shown that SKG can be combined with authenticated encryption (AE) schemes [9,10] in order to overcome trivial man-in-the-middle (MiM) attacks, similarly to known MiM attacks on unauthenticated Diffie–Hellman schemes.

The success of the SKG scheme relies on the reciprocity and variability of wireless channels. On the one hand, the reciprocity property allows both Alice and Bob to measure an identical channel impulse response during the coherence time of the channel [11,12,13], while on the other hand, the variability property of the wireless channel directly affects the key generation rates [14,15,16,17].

However, the exchange of pilots during the channel estimation phase between Alice and Bob could allow an adversary (Mallory) to estimate the channels Alice–Mallory and Bob–Mallory. Having this information, Mallory could inject suitably precoded signals during the SKG process and could potentially control a significant part of the reconciled sequence while remaining undetected. To overcome this, instead of transmitting publicly known pilot signals, we propose a two-way randomized pilot transmission between Alice and Bob. An earlier work studied this problem for an orthogonal frequency-division multiplexing (OFDM) system [18]. Here, we investigate the scenario of a multiple-input–multiple-output (MIMO) system. We prove that if Mallory has one extra antenna with respect to Alice and Bob, she could always launch an injection attack. Next, through theoretical analysis, we show that the proposed pilot randomization scheme successfully reduces an injection attack to a less harmful uncorrelated jamming attack, ensuring that the extracted key bits are secret from both active and passive adversaries.

In the second part of this paper, we delve deeper into jamming attacks over MIMO systems. In particular, we focus on denial of service (DoS) in the form of reactive jamming. We derive the optimal strategies for both the attacker and the legitimate users. Through numerical evaluation, we demonstrate that, depending on their capabilities, reactive jammers could provoke legitimate users to transmit at full power in order to achieve a positive SKG rate.

## 2. System Model

In this work, we consider a time-division duplex MIMO (TDD–MIMO) system consisting of two legitimate nodes and an active adversary, namely, Alice, Bob, and Mallory, respectively. On the one hand, Alice and Bob are generating secret keys using the wireless SKG procedure, while on the other hand, Mallory performs an injection attack on the MIMO links Mallory–Alice and Mallory–Bob. The number of antennas at Alice $N_{A}$ and Bob $N_{B}$ are assumed to be equal, i.e., $N_{A}=N_{B}=N$. To better illustrate the considered scenario, we give a brief overview of the SKG procedure, and show how an injection attack could affect the process.

### 2.1. Secret Key Generation from Fading Coefficients

As illustrated in Figure 1, the standard SKG procedure consists of three phases [19]: (1) advantage distillation: the legitimate nodes exchange pilot signals, each using *N* transmit and *N* receive antenna elements, in order to estimate their reciprocal channel state information (CSI).
(1)$$\begin{array}{cc} \; & z_{A}=Hx+n_{A} \end{array}$$
(2)$$\begin{array}{cc} \; & z_{B}=H^{T}x+n_{B}, \end{array}$$
where H represents the channel matrix of size $N_{r}\times N_{t}=N\times N$ such that its (i,j) entry represents the channel linking the *i*-th receive antenna, and the *j*-th transmit antenna, z represents the received vector of length $N_{r}$, x denotes the transmitted vector consisting of $N_{t}=N_{r}=N$ elements, $n_{A}$ and $n_{B}$ are the received noise vectors at Alice and Bob, each of length $N_{r}$, respectively. Note that, due to the reciprocity of the wireless channel, Alice and Bob observe H and $H^{T}$, respectively. To conclude this step, $z_{A}$ and $z_{B}$ are passed through suitable quantizers [20], generating binary vectors $r_{A}$ and $r_{B}$, respectively; (2) information reconciliation: discrepancies, due to imperfect channel estimation in the quantizer local outputs, are reconciled through a public exchange of helper data $s_{A}$ (see Figure 1), e.g., by using Slepian–Wolf reconciliation techniques [10,21]; (3) privacy amplification: the legitimate nodes apply universal hash functions to the reconciled information $r_{A}$ and obtain key k. This step ensures that the generated key k is uniformly distributed and completely unpredictable by an adversary.

During the process above, an eavesdropping adversary could obtain channel observations, given as follows:(3)$$\begin{array}{cc} \; & z_{AM}=H_{AM}x+n_{AM}, \end{array}$$(4)$$\begin{array}{cc} \; & z_{BM}=H_{BM}x+n_{BM}, \end{array}$$
where the channel matrices in the links Alice–Mallory and Bob–Mallory are denoted by $H_{AM}$ and by $H_{BM}$, respectively, while the received noise vectors are demoted by $n_{AM}$ and $n_{BM}$. Afterward, the SKG capacity between Alice and Bob is expressed as the conditional mutual information between the observations of Alice, Bob, and Mallory.
(5)$$I(z_{A};z_{B}|z_{AM},z_{BM}).$$

### 2.2. Injection Attacks during SKG

One of the most critical threats to the SKG model, given in Figure 1, is MiM in the form of an injection attack [11,22,23]. The main components of the injection attack are captured in Figure 2. While, the legitimate nodes Alice and Bob exchange pilot signals during the advantage distillation phase, Mallory injects signals p. Based on the results in [22], we assume that Mallory has perfect knowledge of the channel vectors in the MIMO links Mallory–Alice, $H_{MA}=H_{AM}^{T}$ and Mallory–Bob, $H_{MB}=H_{BM}^{T}$. This is a reasonable assumption since Mallory can estimate the channel vectors while Alice and Bob exchange pilot signals, as long as the channel’s coherence time is respected (a plausible scenario in slow-fading, low-mobility environments). Finally, Mallory chooses the vector p such that the same signal is “injected“ at both Alice and Bob, i.e., $H_{MA}p=H_{MB}p$.

## 3. Analysis of Injection Attacks in MIMO SKG

In this section, we first prove that if Mallory has one extra antenna, with respect to Alice and Bob, she could always launch an injection attack. Next, we propose a pilot randomization scheme and show that when employed, legitimate users could successfully reduce the attack to a jamming attack.

**Lemma** **1.**
*While Alice and Bob perform advantage distillation using N antennas, Mallory could always launch an injection attack, as long as she has at least N+1 antennas.*


**Proof.** The precoding vector of Mallory p of size (N+1)×1 is represented as
(6)$$p=\left[\begin{array}{c} p_{1}\\ \vdots \\ p_{N+1} \end{array}\right].$$The channel matrices $H_{MA}$ and $H_{MB}$ have size N×(N+1), such that
(7)$$H_{MA}=\left[\begin{array}{ccc} H_{{MA}_{1,1}} & \cdots & H_{{MA}_{1,N+1}}\\ \vdots & \cdots & \vdots \\ H_{{MA}_{N,1}} & \cdots & H_{{MA}_{N,N+1}} \end{array}\right],$$
and
(8)$$H_{MB}=\left[\begin{array}{ccc} H_{{MB}_{1,1}} & \cdots & H_{{MB}_{1,N+1}}\\ \vdots & \cdots & \vdots \\ H_{{MB}_{N,1}} & \cdots & H_{{MB}_{N,N+1}} \end{array}\right].$$Next, we can represent the equation
(9)$$H_{MA}p=H_{MB}p,$$
as
(10)$$(H_{MA}-H_{MB})p=0,$$
where $H_{M}=H_{MA}-H_{MB}$ is equal to:
(11)$$H_{M}=\left[\begin{array}{ccc} H_{MA_{1,1}}-H_{MB_{1,1}} & \cdots & H_{MA_{1,N+1}}-H_{MB_{1,N+1}}\\ \vdots & \cdots & \vdots \\ H_{MA_{N,1}}-H_{MB_{N,1}} & \cdots & H_{MA_{N,N+1}}-H_{MB_{N,N+1}} \end{array}\right].$$Given the above, Equation (Equation 10) can be rewritten as $H_{M}p=0$, where $H_{M}$ is given in Equation (Equation 11). The equality $H_{M}p=0$ is equivalent to solving the following linear system of equations:
(12)$$\left\{\begin{array}{ccc} H_{M_{1,1}}p_{1}+H_{M_{1,2}}p_{2}+\cdots +H_{M_{1,N+1}}p_{N+1} & = & 0\\ \;\;\;\;\;\vdots \\ H_{M_{N,1}}p_{1}+H_{M_{N,2}}p_{2}+\cdots +H_{M_{N,N+1}}p_{N+1} & = & 0. \end{array}\right.$$Due to the fact that Mallory has an additional degree of freedom (one extra antenna), as compared to Alice and Bob, she can treat one of the elements in p as a constant and solve for the others in terms of it. Based on this, we let $p_{N+1}$ be a constant and rewrite the system in (Equation 12) as
(13)$$\left\{\begin{array}{ccc} H_{M_{1,1}}p_{1}+H_{M_{1,2}}p_{2}+\cdots +H_{M_{1,N}}p_{N} & = & -H_{M_{1,N+1}}p_{N+1}\\ \;\;\;\;\;\vdots \\ H_{M_{N,1}}p_{1}+H_{M_{N,2}}p_{2}+\cdots +H_{M_{N,N}}p_{N} & = & -H_{M_{N,N+1}}p_{N+1}. \end{array}\right.$$The system of equations in (Equation 13) can be represented as Ax=b, where the N×N matrix A is the N×N matrix containing the first *N* lines and *N* columns of $H_{M}$, $x={\left(p_{1},p_{2},\cdots ,p_{N}\right)}^{T}$, and b contains the right-hand side of the system, i.e., $b={\left(-H_{M_{1,N+1}}p_{N+1},\cdots ,-H_{M_{N,N+1}}p_{N+1}\right)}^{T}$. Finally, since det(A)≠0 almost surely, (i.e., under the assumptions in Section 2, det(A) is a continuous random variable, hence det(A)≠0 with probability 1) and therefore the system’s solution is unique and given by
(14)$${\left(p_{1},p_{2},\cdots ,p_{N}\right)}^{T}=A^{-1}b.$$Note that if Mallory has the same number of antennas as Alice and Bob, she will not have one extra degree of freedom and the transition from the system in Equation (Equation 12) to the system in Equation (Equation 13) would not be possible. However, as shown here, if Mallory has one extra antenna, with respect to Alice and Bob, she can treat one of the elements in p as constant, which allows her to find the rest of the elements as in Equation (Equation 14). This concludes the proof of Lemma 1. □

Based on Lemma 1, the observations of Alice and Bob are now given by
(15)$$\begin{array}{c} z_{A}=Hx+w+n_{A} \end{array}$$
(16)$$\begin{array}{c} z_{B}=H^{T}x+w+n_{B}, \end{array}$$
where $w=H_{MA}p=H_{MB}p$ denotes the observed injected signals at Alice and Bob, which are identical due to the precoding vector p. By injecting w, Mallory controls the secret key rate, which is now upper bounded by [18,24]
(17)$$L\leq I(z_{A},z_{B};w).$$

### Pilot Randomization as a Countermeasure to Injection Attacks

It has been shown that a countermeasure to injection attacks can be built by randomizing the pilot sequence exchanged between Alice and Bob [18,23,24]. In this work, we propose a MIMO pilot randomization scheme in which pilots are drawn from a (scaled) QPSK modulation. Specifically, Alice and Bob do not transmit the same pilot signal x; instead, they transmit independent, random pilot signals x and y drawn from i.i.d. zero-mean discrete uniform distributions in which the individual elements of the vectors have probability mass functions as $U(\{\pm r\pm jr\},\cdots ,\{\pm r\pm jr\}),\;\mathrm{where}\;j=\sqrt{-1},r=\sqrt{P/2}$, so that $E\left[x\right]=E\left[y\right]={\left(0,\cdots ,0\right)}^{T}$, ${\left(E\left[|x_{1}{|}^{2}\right],\cdots ,E\left[|x_{N}{|}^{2}\right]\right)}^{T}={\left(E\left[|y_{1}{|}^{2}\right],\cdots ,E\left[|y_{N}{|}^{2}\right]\right)}^{T}={\left(P,\cdots ,P\right)}^{T}$ and ${\left(E\left[x_{1}y_{1}\right],\cdots ,E\left[x_{N}y_{N}\right]\right)}^{T}={\left(0,\cdots ,0\right)}^{T}$, i.e., the pilots are randomly chosen QPSK signals. Given that Alice’s and Bob’s observation $z_{A}$ and $z_{B}$ are modified as
(18)$$\begin{array}{cc} \; & z_{A}=Hy+w+n_{A}, \end{array}$$
(19)$$\begin{array}{cc} \; & z_{B}=H^{T}x+w+n_{B}. \end{array}$$

Finally, to generate shared randomness, Alice and Bob post-multiply $z_{A}$ and $z_{B}$ by their own randomized pilot signals, such as ${\tilde{z}}_{A}=x^{T}z_{A}$ and ${\tilde{z}}_{B}=y^{T}z_{B}$ (unobservable by Mallory). Given this, the modified observations are expressed as
(20)$$\begin{array}{ccc} {\tilde{z}}_{A} & = & x^{T}Hy+x^{T}w+x^{T}n_{A}, \end{array}$$
(21)$$\begin{array}{ccc} {\tilde{z}}_{B} & = & y^{T}H^{T}x+y^{T}w+y^{T}n_{B}, \end{array}$$
where the shared randomness between Alice and Bob is now represented by $x^{T}Hy=xH^{T}y^{T}$. Furthermore, the independence of x and y ensures the following:(22)$$L\leq I\left({\tilde{z}}_{A},{\tilde{z}}_{B};w\right)=0.$$

## 4. Jamming Attacks on SKG

In this section, we focus on reactive jamming attacks in SKG systems and examine the scenario in which Mallory reactively jams Alice (note that the scenario in which Mallory jams Bob is identical). A reactive jamming attack is an intelligent approach in which the jammer initially senses the spectrum and jams only if a transmission is detected. Due to the difficulty to be detected, reactive jamming attacks are considered to be a great threat to legitimate transmission [25,26]. Next, we assume that Alice and Bob perform SKG in a TDD–MIMO system with a spatially uncorrelated channel. It has been proven that the optimal power strategy for Alice and Bob in this scenario is to employ equal power distribution [27], which is also assumed for this study, i.e.,
(23)$${\left(E\left[|x_{1}{|}^{2}\right],\cdots ,E\left[|x_{N}{|}^{2}\right]\right)}^{T}={\left(p,\cdots ,p\right)}^{T}\;\mathrm{with}\;p\in \left[0,P\right].$$

In the following, we assume that Mallory has *N* antennas, and as a reactive jammer, she senses the spectrum and jams in the link Mallory–Alice only if she detects a power greater than a certain threshold $p_{\mathrm{th}}$. Thus, instead of considering Mallory’s power allocation matrix, we work with the sum jamming power for all antennas, which can be represented as a power allocation vector $\underline{\gamma }=\left(\gamma_{1},\cdots ,\gamma_{N}\right)$. By denoting the available jamming power by NΓ, the following short-term power constraint is considered:(24)$$\underline{\gamma }\in R_{+}^{N},\;\sum_{i=1}^{N}\gamma_{i}\leq N\Gamma .$$

Assuming that H is uncorrelated with $H_{AM},H_{BM}$ and that all channel matrices have independent and identically distributed elements that are drawn from circularly symmetric zero-mean Gaussian distributions of variances $\sigma^{2}$ and $\sigma_{J}^{2}$, respectively, then the SKG capacity can be expressed as [27]
(25)$$C_{K}\left(p,\underline{\gamma }\right)=N\sum_{i=1}^{N}\log\left(1+\frac{p\sigma^{2}}{2\left(1+\gamma_{i}\sigma_{J}^{2}\right)+\frac{{\left(1+\gamma_{i}\sigma_{J}^{2}\right)}^{2}}{p\sigma^{2}}}\right).$$

### 4.1. Optimal Power Allocation Strategies

In the following, we take a game-theoretic approach in order to evaluate the optimal strategies of Alice, Bob and Mallory. Throughout the following Alice and Bob’s common objective is to maximize $C_{K}\left(p,\underline{\gamma }\right)$ with respect to (w.r.t.) *p*, while Mallory wants to minimize $C_{K}\left(p,\underline{\gamma }\right)$ w.r.t. $\underline{\gamma }$. Due to the reversed objectives, we formulated a noncooperative zero-sum game, which studies the strategic interaction between the legitimate users and the jammer: $G=(\left\{L,J\right\},\left\{A_{L},A_{J}\left(p\right)\right\},C_{K}\left(p,\underline{\gamma }\right))$. The game G has three components: (i) there are two players, namely, *L*, denoting the legitimate users (Alice and Bob act as a single player), and *J* being the jammer (Mallory); (ii) player *L* has a set of possible actions $A_{L}=\left[0,P\right]$, while player *J*’s set of actions is
(26)$$A_{J}\left(p\right)\;=\;\left\{\begin{array}{cc} \{(0,\cdots ,0)\}, & \mathrm{if}\;p\leq p_{\mathrm{th}},\\ \left\{\underline{\gamma }\in R_{+}^{N}|\sum_{i=1}^{N}\gamma_{i}\leq N\Gamma \right\}, & \mathrm{if}\;p>p_{\mathrm{th}}. \end{array}\right.$$

Lastly, $C_{K}\left(p,\underline{\gamma }\right)$ denotes the payoff function of player *L*.

Given the fact that player *J* is a reactive jammer, i.e, first observes the transmit power of player *L* and subsequently chooses a strategy, we study a hierarchical game in which player *L* is the leader, and player *J* is the follower. In this game, the solution is the Stackelberg equilibrium (SE)—rather than Nash—and it is defined as a strategy profile $\left(p^{\mathrm{SE}},{\underline{\gamma }}^{\mathrm{SE}}\right)$ where player *L* chooses their optimal strategy first, by anticipating the strategic reaction of player *J* (i.e., its best response). This is expressed as:(27)$$\begin{array}{cc} \; & p^{\mathrm{SE}}≜\underset{p\in A_{L}}{\arg\max}C_{K}\left(p,{\underline{\gamma }}^{*}\left(p\right)\right),\;\mathrm{and}\;{\underline{\gamma }}^{\mathrm{SE}}≜{\underline{\gamma }}^{*}\left(p^{\mathrm{SE}}\right), \end{array}$$
where ${\underline{\gamma }}^{*}\left(p\right)$ defines the best response (BR) of player *J* to any strategy $p\in A_{L}$ chosen by player *L*, and it is defined as follows:(28)$${\underline{\gamma }}^{*}\left(p\right)≜\underset{\underline{\gamma }\in A_{J}\left(p\right)}{\arg\min}C_{K}\left(p,\underline{\gamma }\right).$$

Finally, based on the detection capabilities at player *L*, two scenarios are considered: (i) when the detection threshold $p_{\mathrm{th}}$ is fixed (defined by the sensing capability of Mallory’s receiver); (ii) when $p_{\mathrm{th}}$ is part of player *L*’s strategy and could vary.

### 4.2. Stackelberg Equilibrium with Fixed Detection Threshold

In this section, we evaluate SE, when player *J*’s detection threshold $p_{\mathrm{th}}$ is predefined and constant. Note that the case $P\leq p_{\mathrm{th}}$ is trivial as ${\underline{\gamma }}^{\mathrm{SE}}=\left(0,\cdots ,0\right)$, and the legitimate users will optimally use their maximum available power, i.e., $\left(p^{\mathrm{SE}}=P\right)$. Indeed, due to the poorly chosen threshold $p_{\mathrm{th}}$ or low sensing capabilities of Mallory, the legitimate transmission will not be detected and therefore will not be jammed. In the following, we assume that $P>p_{\mathrm{th}}$.

**Lemma** **2.**
*The BR of player J for any $p\in A_{L}$ chosen by player L defined in (Equation 28) is the uniform power allocation, given as*
(29)$${\underline{\gamma }}^{*}\left(p\right)≜\left\{\begin{array}{cc} (\Gamma ,\cdots ,\Gamma ), & if\;p>p_{\mathrm{th}},\\ (0,\cdots ,0), & if\;p\leq p_{\mathrm{th}}. \end{array}\right.$$


**Proof.** Note that $C_{K}\left(p,\gamma_{i}\right)$ is a monotonically decreasing convex function w.r.t $\gamma_{i},\;i=1,\ldots ,N$ for any p>0. Based on the principles of convexity in order to minimize $C_{K}$, Mallory has to transmit with full power from all antennas. The detailed proof can be found in [18]. □

Based on the result from Lemma 1, the SKG rate can have the following two forms:(30)$$C_{K}\left(p,{\underline{\gamma }}^{*}\left(p\right)\right)=\left\{\begin{array}{cc} C_{K}\left(p,\left(0,\cdots ,0\right)\right), & \mathrm{if}\;p\leq p_{\mathrm{th}},\\ C_{K}\left(p,\left(\Gamma ,\cdots ,\Gamma \right)\right), & \mathrm{if}\;p>p_{\mathrm{th}}, \end{array}\right.$$
which simplifies the players’ options.

**Theorem** **1.**
*Depending on their available power P for SKG, Alice and Bob will either transmit at P or $p_{\mathrm{th}}$. The SE point of the game is unique when $P\neq p_{\mathrm{th}}\left(\Gamma \sigma_{J}^{2}+1\right)$ and is given by*
(31)$$\left(p^{\mathrm{SE}},{\underline{\gamma }}^{\mathrm{SE}}\right)\;\;=\;\;\left\{\;\;\;\;\begin{array}{cc} \{\left(p_{\mathrm{th}},\left(0,\cdots ,0\right)\right)\}, & \;\;\;\;\mathrm{if}\;P<p_{\mathrm{th}}\left(\sigma_{J}^{2}\Gamma \;+\;1\right),\\ \{(P,(\Gamma ,\cdots ,\Gamma ))\}, & \;\;\;\;\mathrm{if}\;P>p_{\mathrm{th}}\left(\sigma_{J}^{2}\Gamma \;+\;1\right). \end{array}\right.$$

*When $P=p_{\mathrm{th}}\left(\sigma_{J}^{2}\Gamma +1\right)$, the game G has two SEs: $\left(p^{\mathrm{SE}},{\underline{\gamma }}^{\mathrm{SE}}\right)\in \left\{\left(p_{\mathrm{th}},\left(0,\cdots ,0\right)\right),\left(P,\left(\Gamma ,\cdots ,\Gamma \right)\right)\right\}$.*


**Proof.** Given the BR of player *J* defined in (Equation 29), the legitimate users want to identify their optimal $p\in A_{L}$ that maximizes
(32)$$C_{K}\left(p,{\underline{\gamma }}^{*}\left(p\right)\right)=\left\{\begin{array}{cc} C_{K}\left(p,\left(0,\cdots ,0\right)\right), & \mathrm{if}\;p\leq p_{\mathrm{th}},\\ C_{K}\left(p,\left(\Gamma ,\cdots ,\Gamma \right)\right), & \mathrm{if}\;p>p_{\mathrm{th}}, \end{array}\right.$$Given the fact that $C_{K}\left(p,\underline{\gamma }\right)$ is monotonically increasing with *p* for fixed $\underline{\gamma }$, two cases are distinguished: (a) $p\in [0,p_{\mathrm{th}}]$, (b) $p\in (p_{\mathrm{th}},P]$. The optimal *p* in each case is given by(a) $\underset{p\in [0,p_{\mathrm{th}}]}{\arg\max}\;C_{K}\left(p,{\underline{\gamma }}^{*}\left(p\right)\right)=\underset{p\in [0,p_{\mathrm{th}}]}{\arg\max}\;C_{K}(p,\left(0,\cdots ,0\right)=p_{\mathrm{th}},$(b) $\underset{p\in (p_{\mathrm{th}},P]}{\arg\max}\;C_{K}\left(p,{\underline{\gamma }}^{*}\left(p\right)\right)=\underset{p\in (p_{\mathrm{th}},P]}{\arg\max}\;C_{K}(p,\left(\Gamma ,\cdots ,\Gamma \right)=P.$From (a) and (b), it can be concluded that the overall solution is $p^{\mathrm{SE}}=$
$$\underset{p\in A_{L}}{\arg\max}\;C_{K}\left(p,{\underline{\gamma }}^{*}\left(p\right)\right)=\left\{\begin{array}{cc} \;\;p_{\mathrm{th}},\;\;\; & \mathrm{if}\;C_{K}\left(P,\Gamma \right)<C_{K}\left(p_{\mathrm{th}},0\right),\\ \;\;P,\;\;\; & \mathrm{if}\;C_{K}\left(P,\Gamma \right)>C_{K}\left(p_{\mathrm{th}},0\right),\\ \;\;\{p_{\mathrm{th}},P\},\;\;\; & \mathrm{if}\;C_{K}\left(P,\Gamma \right)=C_{K}\left(p_{\mathrm{th}},0\right). \end{array}\right.$$To simplify the above possibilities, we focus on the case when the utility function $C_{K}\left(P,\Gamma \right)$, i.e., being detected and jammed, equals the utility function when player *L* is transmitting at threshold $p_{\mathrm{th}}$ (player *J* is silent), i.e., $C_{K}\left(P,\Gamma \right)=C_{K}\left(p_{\mathrm{th}},0\right)$. Using this equality, by substituting appropriately into (Equation 25), we obtain a quadratic equation in *P*.
$$P^{2}\;\left(2\sigma^{2}p_{\mathrm{th}}\;+\;1\right)\;-\;P\left(2{p_{\mathrm{th}}}^{2}\sigma^{2}\;+\;2\sigma_{J}^{2}\Gamma {p_{\mathrm{th}}}^{2}\sigma^{2}\right)-{\left(1\;+\;\sigma_{J}^{2}\Gamma \right)}^{2}{p_{\mathrm{th}}}^{2}\;=\;0.$$Note that Equation (33) has a unique positive root equal to $p_{\mathrm{th}}\left(\sigma_{J}^{2}\Gamma +1\right)$. Furthermore, due to the fact that the leading coefficient of (33): $(2\sigma^{2}p_{\mathrm{th}}+1)\geq 0$ and P>0, we can state that the inequalities $C_{K}\left(P,\Gamma \right)>C_{K}\left(p_{\mathrm{th}},0\right)$ and $C_{K}\left(P,\Gamma \right)<C_{K}\left(p_{\mathrm{th}},0\right)$ are equivalent to $P>p_{\mathrm{th}}\left(\sigma_{J}^{2}\Gamma +1\right)$ and $P<p_{\mathrm{th}}\left(\sigma_{J}^{2}\Gamma +1\right)$, respectively. □

A numerical evaluation of the SKG rate is presented in Figure 3. The parameters used are N=10, $p_{\mathrm{th}}=2$, Γ=3, and $\sigma^{2}=\sigma_{J}^{2}=1$. Figure 3 compares the achievable SKG rates of the SE strategy, i.e., $p=p^{SE}$ with the two alternative strategies, i.e., p=P or $p=p_{\mathrm{th}}$. It can be seen that if player *L* deviates from the SE point the achievable SKG rate can decrease by up to 40%.

### 4.3. Stackelberg Equilibrium with Strategic $p_{\mathrm{th}}$

Finally, we investigate the case when Mallory could optimally adjust $p_{\mathrm{th}}$ and show how her choice impacts Alice’s and Bob’s strategies. Allowing $p_{\mathrm{th}}$ to vary modifies the game under study as follows $\hat{G}=\left(\left\{L,J\right\},\left\{A_{L},{\hat{A}}_{J}\left(p\right)\right\},C_{K}\left(p,\underline{\gamma },p_{\mathrm{th}}\right)\right)$, where
(33)$${\hat{A}}_{J}\left(p\right)≜\left\{\begin{array}{cc} \;\;\;\{\left(\left(0,\cdots ,0\right),p_{\mathrm{th}}\right),\;p_{\mathrm{th}}\geq 0\},\;\;\; & \mathrm{if}\;p_{\mathrm{th}}\geq p,\\ \;\;\;\left\{\left(\underline{\gamma },p_{\mathrm{th}}\right)\in R_{+}^{N}\;|\;\sum_{i=1}^{N}\gamma_{i}\leq N\Gamma \right\},\;\;\;\; & \mathrm{if}\;p_{\mathrm{th}}<p. \end{array}\right.$$

The BR of the jammer can then be defined as
(34)$$\left({\hat{\underline{\gamma }}}^{*}\left(p\right),{\hat{p_{\mathrm{th}}}}^{*}\left(p\right)\right)≜\underset{\left(\underline{\gamma },p_{\mathrm{th}}\right)\in {\hat{A}}_{J}\left(p\right)}{\arg\min}C_{K}\left(p,\underline{\gamma },p_{\mathrm{th}}\right).$$

**Lemma** **3.**
*Mallory’s BR in this scenario is a set of strategies as follows:*
(35)$$\begin{array}{c} \left({\hat{\underline{\gamma }}}^{*}\left(p\right),{\hat{p_{\mathrm{th}}}}^{*}\left(p\right)\right)\in \left\{\;\left(\left(\Gamma ,\cdots ,\Gamma \right)\epsilon \right),\;\epsilon \in \left[0,p\right)\right\}. \end{array}$$


**Proof.** The problem that the jammer wants to solve is $\underset{\left(\underline{\gamma },p_{\mathrm{th}}\right)\in {\hat{A}}_{J}\left(p\right)}{\min\;}C_{K}\left(p,\underline{\gamma },p_{\mathrm{th}}\right),$ which can be split as follows:
(36)$$\begin{array}{c} \min_{p_{\mathrm{th}}\geq 0}\;\;\underset{\underline{\gamma }\in {\hat{A}}_{J}\left(p\right)}{\min\;}C_{K}\left(p,\underline{\gamma }\left(p\right),p_{\mathrm{th}}\right). \end{array}$$The solution of the inner minimization is known from (Equation 29). For the outer problem, we have to find the optimal $p_{\mathrm{th}}\geq 0$ that minimizes $C_{K}\left(p,{\hat{\underline{\gamma }}}^{*}\left(p\right),p_{\mathrm{th}}\right)$. Given that
(37)$$\min_{p_{\mathrm{th}}\geq 0}C_{K}\left(p,{\hat{\underline{\gamma }}}^{*}\left(p\right),p_{\mathrm{th}}\right)\;\;=\;\;\left\{\begin{array}{cc} C_{K}\left(p,\Gamma ,p_{\mathrm{th}}\right), & \mathrm{if}\;p_{\mathrm{th}}<p,\\ C_{K}\left(p,0,p_{\mathrm{th}}\right), & \mathrm{if}\;p_{\mathrm{th}}\geq p, \end{array}\right.$$
and that $C_{K}\left(p,\Gamma ,p_{\mathrm{th}}\right)<C_{K}\left(p,0,p_{\mathrm{th}}\right)$, player *J* can optimally choose any $p_{\mathrm{th}}$ such that $p_{\mathrm{th}}=\epsilon ,\;\;\forall \epsilon <p$. This allows the jammer to detect any ongoing transmission and to perform a jamming attack. □

**Theorem** **2.**
*The game $\hat{G}$ has an infinite number of SEs as follows:*
(38)$$\left({\hat{p}}^{\mathrm{SE}},{\hat{\underline{\gamma }}}^{\mathrm{SE}},{\hat{p_{\mathrm{th}}}}^{\mathrm{SE}}\right)\in \left\{\;\left(P,\left(\Gamma ,\cdots ,\Gamma \right)\epsilon \right),\;\;\forall \epsilon <P\right\}.$$


**Proof.** Given Mallory’s BR, we evaluate the SE of the game $\hat{G}$. The definition for ${\hat{p}}^{\mathrm{SE}}$ is given as follows:
(39)$${\hat{p}}^{\mathrm{SE}}≜\underset{p\in A_{L}}{\arg}\max C_{K}\left(p,{\hat{\underline{\gamma }}}^{*}\left(p\right),\hat{p_{\mathrm{th}}}{\left(p\right)}^{*}\right).$$Since Mallory will act as in (35), we have
(40)$$C_{K}\left(p,{\hat{\underline{\gamma }}}^{*}\left(p\right),\hat{p_{\mathrm{th}}}{\left(p\right)}^{*}\right)=C_{K}\left(p,\Gamma ,\epsilon \right),\;\forall \epsilon <p,$$
and the fact that $C_{K}\left(p,\Gamma ,\epsilon \right)$ is monotonically increasing with *p* results in ${\hat{p}}^{\mathrm{SE}}=P$. □

Figure 4 illustrates the achievable SKG rate when $p_{\mathrm{th}}$ is part of player *J*’s strategy. As in Figure 3, the parameters are chosen as Γ=3, N=10 and $\sigma_{J}^{2}=1$. It can be seen that due to a strategically chosen threshold from player *J* the legitimate users have no other choice but to transmit at full power $p=P=p^{\mathrm{SE}}$. In fact, if the legitimate users deviate from the SE strategy and transmit with low power $p=p_{\mathrm{th}}$, player *J* could successfully disrupt their SKG process and decrease their achievable SKG rate by up to 97%.

## 5. Conclusions

In this study, injection and reactive jamming attacks were analyzed in MIMO SKG systems. With respect to injection attacks, the study demonstrated that a trivial advantage in the form of one extra antenna allows a MiM to mount such an attack. As a countermeasure, we showed that a pilot randomization scheme can successfully reduce injection attacks to jamming attacks. With respect to jamming attacks, using a game-theoretic approach, we showed that an intelligent reactive jammer should optimally jam with full power when a transmission is sensed. Finally, by strategically choosing her jamming threshold, i.e., just below the power level used by the legitimate users, Mallory could perform a much more effective attack. In fact, our theoretical analysis suggests that in this case, Alice and Bob have no choice but to use their full power available for SKG. An important topic for further research in this area is an examination of these initial findings in practical scenarios.

## Figures and Tables

**Figure 1 entropy-23-00960-f001:**
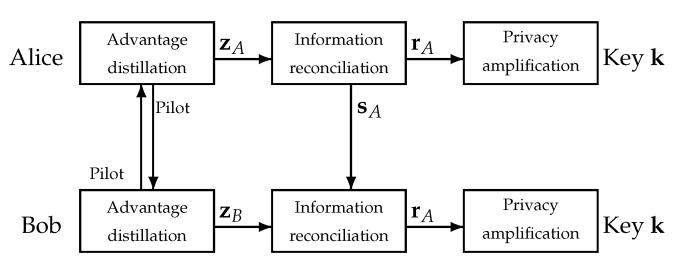
Secret key generation process between Alice and Bob.

**Figure 2 entropy-23-00960-f002:**
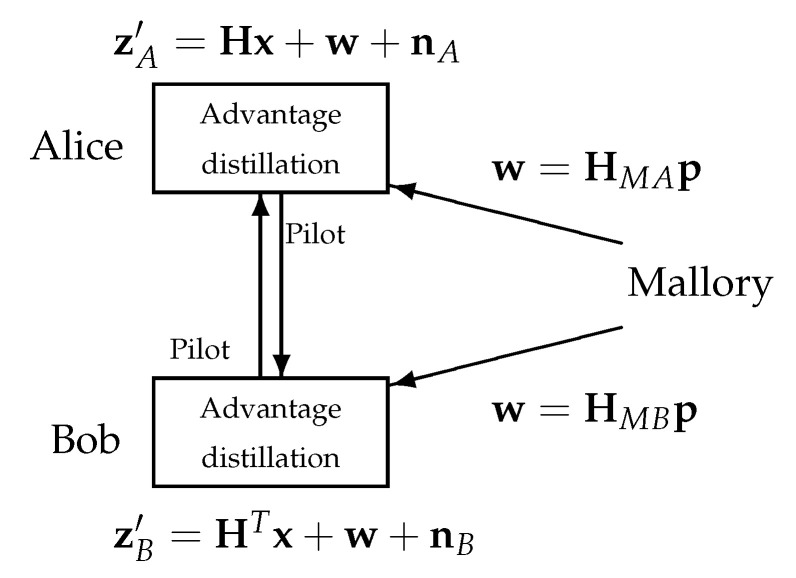
Injection attack performed by Mallory: While Alice and Bob exchange pilot signals x over a Rayleigh fading channel with realization H, Mallory injects a signal p such that the received signals at both Alice and Bob coincide $w=H_{MA}p=H_{MB}p$.

**Figure 3 entropy-23-00960-f003:**
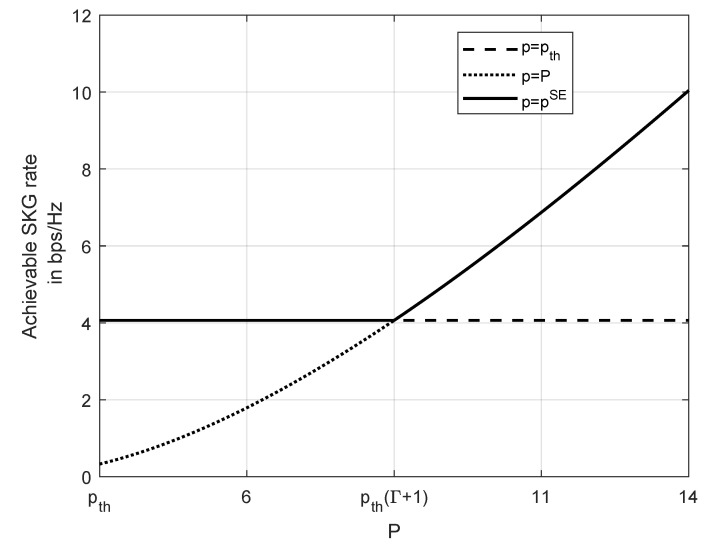
SE policy, compared to always transmitting with either full power or with $p_{\mathrm{th}}$. Used parameters $p_{\mathrm{th}}=2,\Gamma =3,N=10,\sigma^{2}=\sigma_{J}^{2}=1$.

**Figure 4 entropy-23-00960-f004:**
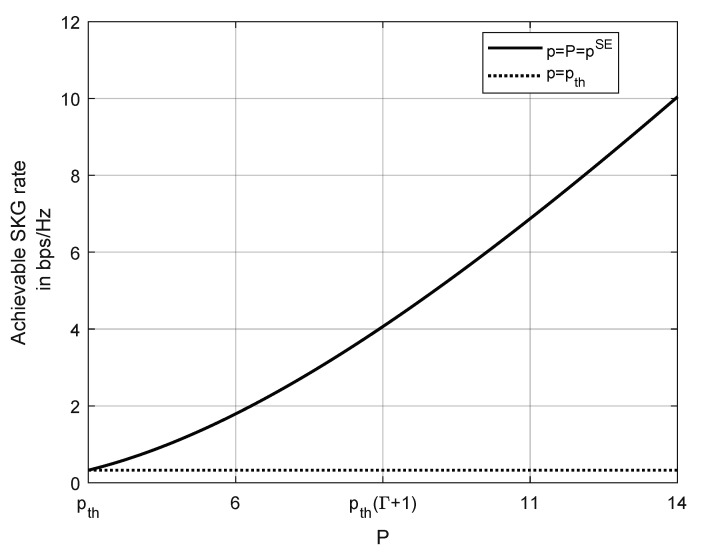
The effect to the SE policy when $p_{\mathrm{th}}$ is part of player *J* strategy. Comparison of the achievable SKG rate when player *L* chooses $p=p^{\mathrm{SE}}$ with the case when transmitting with power $p_{\mathrm{th}}$. Used parameters $\Gamma =3,N=10,\sigma^{2}=\sigma_{J}^{2}=1$.

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
