# Peer review of "Protecting Physical Layer Secret Key Generation from Active Attacks"

_entropy, 2021, doi:10.3390/e23080960_

Round 1

Reviewer 1 Report

In this paper, the authors analyzed injection and reactive jamming attacks in MIMO secret key generation systems, and proposed a scheme to reduce injection attacks to less harmful jamming attacks and derived optimal power allocation policies for both legitimate user and malicious adversaries.

In general, the paper is well written with clear formulation and sound analysis. On the other hand, the work reported in this paper is mainly based on theoretical analysis with very limited numerical results. More evaluation of the proposed scheme in a realistic scenario using either prototype testing or simulation would be helpful to show the effectiveness of the work in practice.  

Reviewer 2 Report

This paper presents new results on secret key generation for MIMO systems.  The authors analyze man-in-the-middle attacks, and show when they are possible.  They then propose a randomization scheme that mitigates the effect of the man-in-the-middle attack, reducing it to a jamming-only attack.  The results are interesting and thorough.  The paper is well structured.

My only complaint with the paper is the presence of a number of typos.  I'm attaching a pdf with typos circled in red.  There may be others that I did not notice.  I suggest that the authors also perform a thorough check for additional ones.

In my opinion, this paper should be published after addressing these minor issues.
